# Auxin production in the endosperm drives seed coat development in *Arabidopsis*

Duarte D Figueiredo, Rita A Batista, Pawel J Roszak[†], Lars Hennig, Claudia Köhler*

Department of Plant Biology, Uppsala BioCenter, Swedish University of Agricultural Sciences and Linnean Center for Plant Biology, Uppsala, Sweden

**Abstract** In flowering plants, seed development is initiated by the fusion of the maternal egg and central cells with two paternal sperm cells, leading to the formation of embryo and endosperm, respectively. The fertilization products are surrounded by the maternally derived seed coat, whose development prior to fertilization is blocked by epigenetic regulators belonging to the Polycomb Group (PcG) protein family. Here we show that fertilization of the central cell results in the production of auxin and most likely its export to the maternal tissues, which drives seed coat development by removing PcG function. We furthermore show that mutants for the MADS-box transcription factor AGL62 have an impaired transport of auxin from the endosperm to the integuments, which results in seed abortion. We propose that AGL62 regulates auxin transport from the endosperm to the integuments, leading to the removal of the PcG block on seed coat development.

**\*For correspondence:** claudia.
kohler@slu.se

**Present address:** [†]The
Sainsbury Laboratory, University
of Cambridge, Cambridge,
United Kingdom

**Competing interests:** The
authors declare that no
competing interests exist.

**Reviewing editor:** Richard
Amasino, University of
Wisconsin, United States

## Introduction

In flowering plants, fertilization of the two female gametes, egg cell and central cell by the two male sperm cells results in the development of the embryo and the endosperm, a nourishing tissue which supports embryo growth. These two fertilization products are surrounded by the seed coat, a sporophytic tissue of purely maternal origin. While the embryo is the only component of the seed that will form the next generation, it contributes little to seed growth. Instead, the interplay between the endosperm and seed coat determines the final size of the seed. Impaired endosperm proliferation has a negative effect on seed coat development, as is the case in the *Arabidopsis haiku* or *miniseed* mutants (*Garcia et al., 2003*; *Luo et al., 2005*) or in transgenic lines expressing diphtheria toxin A in the endosperm (*Weijers et al., 2003*). And, conversely, mutations that affect seed coat expansion limit endosperm growth, such as *transparent testa glabra 2* (*Garcia et al., 2005*), while mutants with increased integument cell proliferation like *megaintegumenta/auxin responsive factor 2*, result in enlarged seeds with more abundant endosperm (*Schruff et al., 2006*). Furthermore, the absence of the endothelium integument layer results in seed abortion, highlighting the importance of the developing seed coat for the establishment of a viable seed (*Mizzotti et al., 2012*).

Endosperm initiation in *Arabidopsis* is coupled to the production of auxin in the central cell following fertilization (*Figueiredo et al., 2015*); however, the factor initiating seed coat development is yet to be described. Seed coat initiation is dependent on the fertilization of the female gametophyte, but mutants for sporophytically-acting Polycomb Group (PcG) proteins develop a seed coat without fertilization in a dosage-sensitive manner (*Roszak and Köhler, 2011*). PcG proteins assemble into multimeric complexes, of which the Polycomb Repressive Complex 2 (PRC2) represses target loci through the deposition of trimethyl groups on lysine 27 of histone H3 (H3K27me3) (*Mozgova et al., 2015*). Therefore, the initiation of seed coat development prior to fertilization is inhibited by sporophytically active PRC2 complexes and this block is relieved following fertilization

**eLife digest** The seeds of rice, wheat and other flowering plants store a variety of nutrients, largely in the form of sugars, proteins and oils. These stored reserves provide the main source of calories for humans and livestock all over the world, so they are of major social and economic importance.

Seed development is an intricate process. It begins after male sperm cells fuse with female gametes inside the flower. This leads to the formation of the embryo, which will develop into a new plant, and a structure called the endosperm, which nourishes the growing embryo. A protective seed coat surrounds the embryo and endosperm, which develops from certain parts of the parent flower. In order for the seed to develop successfully, these three components have to communicate so they can coordinate their growth.

Auxin is a key plant hormone that is needed for plants to grow and develop properly and is necessary for the endosperm to form. Previous research has shown that the endosperm is also required to trigger the formation of the seed coat, but the signal that triggers this process has not yet been identified. Figueiredo et al. now address this question in a small flowering plant called *Arabidopsis thaliana*.

The experiments show that the endosperm produces auxin, which acts as a molecular signal for the seed coat to start forming. Exposing unfertilized flowers to auxin caused a seed coat to form even though the endosperm was absent. This suggests that this hormone alone is sufficient to trigger the formation of the seed coat without any other signals. Further analysis revealed that a protein called AGL62 regulates the movement of auxin to the parts of the flower that give rise to the seed coat. In the absence of AGL62, the hormone remains trapped in the endosperm and the seed coat fails to develop. The next step following on from this work is to understand how auxin moves from the endosperm to the parts of the flower that form the seed coat.

through a signal derived from the fertilization products (*Figueiredo and Köhler, 2014*; *Mozgova et al., 2015*). Seed coat development is dependent on the development of the sexual endosperm (*Weijers et al., 2003*; *Ingouff et al., 2006*; *Roszak and Köhler, 2011*), strongly suggesting that the seed coat-initiation signal is generated in the fertilized central cell/endosperm.

Here, we show that seed coat development requires production of auxin in the fertilized central cell/endosperm and most likely the transport of auxin to the ovule integuments in an AGL62-dependent manner. We demonstrate that application of auxin is sufficient to drive seed coat development and that AGL62 regulates the expression of *P-GLYCOPROTEIN 10* (*PGP10*), in the endosperm, which likely functions as an auxin transporter to the sporophytic tissues. Finally, we show that auxin is sufficient to remove the PRC2 block on seed coat development.

## Results

### Auxin and gibberellin signaling are active in the developing seed coat

Seed coat initiation is dependent on the fertilization of the central cell by one of the paternally-contributed sperm cells, but this requirement can be bypassed in mutants of sporophytic PRC2 components that initiate the autonomous development of the seed coat (*Roszak and Köhler, 2011*). In order to identify the signaling pathways that could be involved in seed coat initiation, we generated transcriptome data from non-fertilized wild-type (WT) ovules and *vrn2*/- *emf2*/+ ovules at four days after emasculation (4 DAE), and WT seeds at two days after pollination (2 DAP). In contrast to WT ovules, non-fertilized *vrn2*/- *emf2*/+ ovules initiate autonomous seed coat development (*Roszak and Köhler, 2011*), correlating with the activation of genes involved in auxin and gibberellin (GA) response or signaling that became also activated in fertilized seeds (*Table 1* and *Table 1—source data 1*).

Post-fertilization activation of both auxin and GA signaling was previously shown in *Arabidopsis* seeds (*Dorcey et al., 2009*). In order to test whether this activation was specific to the seed coat we investigated the behavior of auxin and GA reporter lines before and after fertilization (*Figure 1*). To

**Table 1.** Significantly enriched biological processes for genes commonly upregulated in fertilized WT and autonomous *vrn2 emf2/+* seeds, compared to unfertilized WT ovules (p-value<0.05).

| GO-term | p-value | Number of genes | Description |
|---|---|---|---|
| GO:0016043 | 2,12E-10 | 63 | cell organization and biogenesis |
| GO:0006412 | 2,17E-09 | 66 | translation |
| GO:0009058 | 1,46E-07 | 96 | biosynthetic process |
| GO:0042545 | 1,04E-04 | 10 | cell wall modification |
| GO:0009753 | 6,58E-04 | 10 | response to jasmonic acid |
| **GO:0009739** | **1,80E-03** | **8** | **response to gibberellin** |
| GO:0009605 | 1,94E-03 | 16 | response to external stimulus |
| GO:0008361 | 1,98E-03 | 10 | regulation of cell size |
| GO:0009861 | 2,09E-03 | 10 | jasmonic acid and ethylene-dependent systemic resistance |
| GO:0009611 | 2,60E-03 | 12 | response to wounding |
| GO:0016049 | 5,22E-03 | 9 | cell growth |
| GO:0006694 | 7,68E-03 | 4 | steroid biosynthetic process |
| GO:0007155 | 8,49E-03 | 4 | cell adhesion |
| GO:0009723 | 1,02E-02 | 8 | response to ethylene |
| **GO:0009733** | **1,46E-02** | **12** | **response to auxin** |
| GO:0007267 | 1,81E-02 | 3 | cell-cell signaling |
| GO:0007276 | 1,96E-02 | 6 | gamete generation |
| GO:0009698 | 2,51E-02 | 7 | phenylpropanoid metabolic process |
| GO:0009813 | 2,60E-02 | 4 | flavonoid biosynthetic process |
| GO:0009812 | 3,53E-02 | 4 | flavonoid metabolic process |

**Source data 1.** Seed and ovule transcriptome data. This table includes absolute and relative gene expression values for the microarray data of WT vs. *vrn2/- emf2/+* ovules and seeds, and for the mRNAseq data of WT vs. *agl62/+* seeds.

monitor auxin signaling, we investigated expression of the *DR5v2::VENUS* reporter (*Liao et al., 2015*) before and after fertilization. There was no *DR5v2::VENUS* reporter activity in unfertilized ovules (*Figure 1A*), with the exception of a few cells neighboring the vascular bundle and occasionally near the antipodal cells. However, shortly after fertilization VENUS activity was detectable in the sporophytic tissues, where it persisted throughout seed development, with particular incidence in the funiculus and the micropylar region (*Figure 1B–C*). We further substantiated these observations with the use of the R2D2 auxin sensor system (*Liao et al., 2015*). Before fertilization there was a strong DII:VENUS expression in the integuments, indicating very low or absent auxin signaling (*Figure 1D*). However, this signal was quickly depleted following fertilization, as early as the first division of the central cell (*Figure 1E*). Conversely, the auxin-insensitive mDII:Tdtomato remained stable in the seed coat, indicating a stable expression of the *RPS5a* promoter, that drives both reporter genes. These observations indicate a rapid accumulation of auxin in the sporophytic tissues following fertilization of the maternal gametes.

We further analyzed plants expressing the DELLA reporter *RGA::GFP:RGA*, as a marker for active GA signaling (*Dorcey et al., 2009*). Before fertilization we observed a strong GFP signal in the integuments which quickly diminished after fertilization (*Figure 1F–G*), indicating an activation of GA signaling, similarly to what we observed with the auxin reporters.

To test whether auxin and GA act in the same pathway during seed coat development, we treated unpollinated pistils of plants expressing the RGA reporter with the synthetic auxin 2,4-Dichlorophenoxyacetic acid (2,4-D) and observed a removal of the RGA protein, similarly to what happens after fertilization or application of gibberellic acid (GA$_3$; *Figure 1H* and *Figure 1—figure supplement 1*). Conversely, the application of GA$_3$ did not have an influence on the expression of

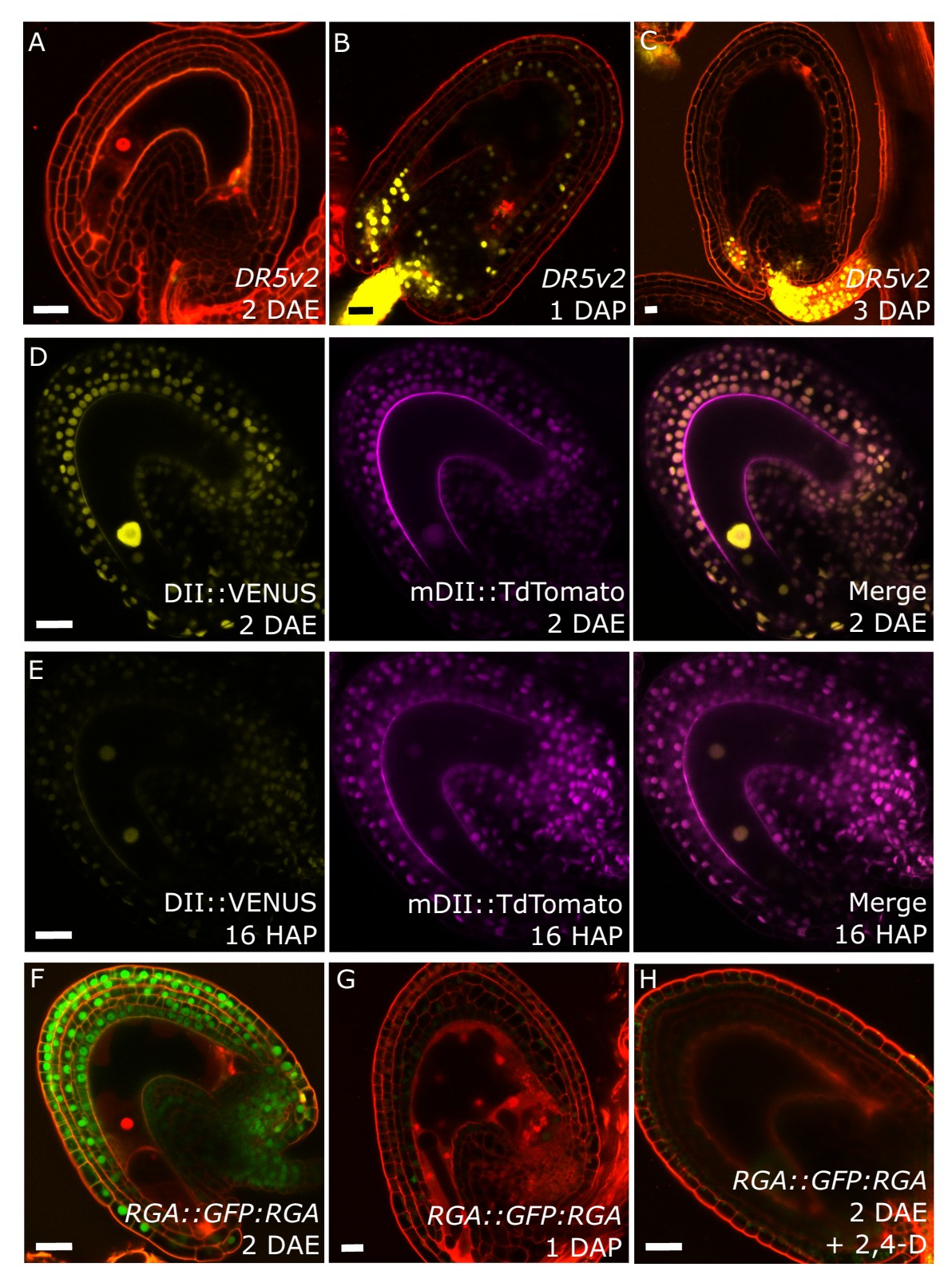

**Figure 1.** Fertilization activates auxin and GA signaling in the seed coat. (**A–C**) *DR5v2::VENUS* expression before (**A**) and one and three days after pollination (**B–C**). (**D–E**) R2D2 auxin sensor before (**D**) and 16 hr after pollination (HAP) (**E**). (**F–H**) *RGA::GFP:RGA* reporter before (**F**), one day after pollination (**G**) and after 2,4-D treatment without pollination (**H**). Bars indicate 20 µm. Red staining is propidium iodide (PI).

*Figure 1 continued on next page*

*Figure 1 continued*

The following figure supplements are available for figure 1:

**Figure supplement 1.** Effect of GA treatments on GA and auxin reporter lines.

**Figure supplement 2.** Ovule size and cell measurements.

the auxin reporters *DR5v2* or R2D2 (*Figure 1—figure supplement 1*). These observations reveal that auxin acts upstream of GA during seed coat development.

## Auxin and GA trigger autonomous seed coat development

Seed coat growth initiating after fertilization is a process driven by cell elongation rather than cell division, since the integument cell number did not change after fertilization (*Figure 1—figure supplement 2*). Given that both auxin and gibberellin signaling is active in the developing seed coat, and that both hormones are known for having a role in cell growth and expansion (*Rayle and Cleland, 1992*; *Cowling and Harberd, 1999*), we tested whether exogenous application of either auxin or GA$_3$ was sufficient to drive seed coat development. We treated unpollinated pistils with either GA$_3$ or 2,4-D and investigated autonomous seed coat development after three days (*Figure 2A–B*). We used vanillin staining as a marker for the seed coat development, which stains the proanthocyanidins produced in the endothelium after fertilization (*Debeaujon et al., 2000*). Mock-treated ovules were either not or only weakly stained with vanillin, while treatments with GA$_3$ or 2,4-D resulted in 30% to 40% of ovules showing strong or very strong staining. Furthermore, the ovules treated with either GA$_3$ or 2,4-D were significantly larger than the mock-treated ones (*Figure 2C*).

We further tested whether over-production of either one of these hormones specifically in the integuments of unfertilized ovules would trigger seed coat development. We raised transgenic plants expressing a GA biosynthesis gene (*GA3ox1*) under the control of the *BANYULS* (*BAN*) promoter, which is active in the innermost integument/seed coat layer (*Figure 2—figure supplement 1*) (*Debeaujon et al., 2003*). We emasculated *BAN::GA3ox1* transgenic plants and performed vanillin staining at 5 DAE to test for initiation of seed coat development. As with the GA$_3$ treatments, we observed an increased number of ovules stained in the transgenic lines when compared to WT plants (*Figure 2D–F*). In order to test whether increased auxin production in the integuments triggers autonomous seed coat development, we analyzed the gain-of-function *yuc6-2D* mutant, in which the auxin biosynthesis gene *YUCCA6* (*YUC6*) is ectopically expressed due to the insertion of a Cauliflower Mosaic Virus 35S (CaMV35S) enhancer sequence in its promotor region (*Kim et al., 2007*). Activation of the *YUC6* gene in *yuc6-2D* is expected to be restricted to sporophytic tissues, as the CaMV35S enhancer is not active in the female gametophyte (*Roszak and Köhler, 2011*). We therefore tested whether increased auxin production in the integuments of *yuc6-2D* plants resulted in the autonomous development of the seed coat, as scored by vanillin staining. Indeed, up to 50% of *yuc6-2D* ovules stained with vanillin, while less than 5% of WT ovules were stained (*Figure 2D*). These observations reveal that both auxin and GA$_3$ are sufficient to trigger seed coat development without fertilization.

## Auxin production is necessary for seed coat development

Based on our findings that ectopic auxin and GA$_3$ are sufficient to trigger the seed coat development and that auxin acts upstream of GA, we hypothesized that auxin may be the post-fertilization signal that triggers seed coat development. This hypothesis is in agreement with our previous observations that auxin is produced in the endosperm after fertilization (*Figueiredo et al., 2015*). In order to test whether auxin produced in the endosperm is necessary for seed coat development, we analyzed mutants deficient either in auxin signaling (*axr1/+ axl-1/-*) or auxin biosynthesis (*wei8/- tar1/- tar2-1/+* and *wei8/- tar1/- tar2-2/+*). The homozygous *axr1/- axl-1/-* mutant is zygotic lethal, thus 25% of seeds from an *axr1/+ axl-1/-* plant will abort (*Dharmasiri et al., 2007*). Nevertheless, those seeds did not differ in size compared to WT seeds at 3 DAP and were only slightly smaller compared to their WT counterparts at 5 DAP, indicating that seed coat development is not impaired (*Figure 3A* and *Figure 3—figure supplement 1*). We thus conclude that auxin signaling in the

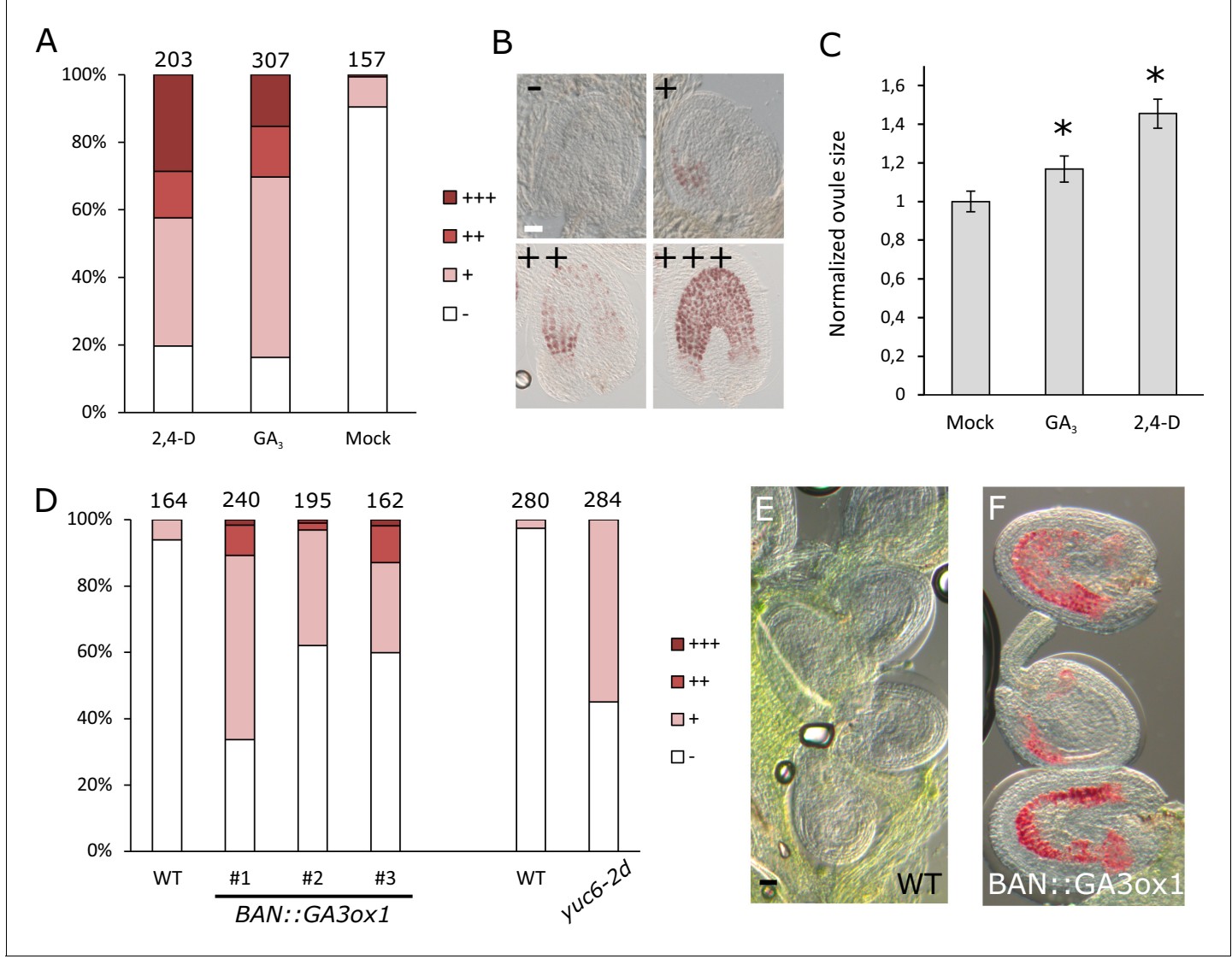

**Figure 2.** GA and auxin are sufficient to drive seed coat development. (A–B) Scoring of vanillin staining of ovules after 2,4-D, GA₃ and mock treatments, as categorized in (B). Numbers on top indicate total ovules counted. Bar indicates 20 µm. (C) Ovule area three days after 2,4-D, GA₃ and mock treatments. Ten ovules were measured per treatment. The size is normalized to the mock-treated ovules. Error bars indicate standard deviation. * Differences are significant for p<0.00001(T-Test). (D) Scoring of vanillin staining in ovules of WT, auxin and GA-overexpressing lines, as categorized in (B). Numbers on top indicate total ovules counted. (E–F) Vanillin-stained WT (E) and *BAN::GA3ox1* (F) ovules at 5 DAE. Bar indicates 50 µm.

The following figure supplement is available for figure 2:

**Figure supplement 1.** Activity of the *BAN* promoter in unfertilized ovules.

fertilization products (endosperm and embryo) is not necessary to initiate seed coat development. In contrast, *wei/tar* mutants affected in auxin biosynthesis showed seeds aborting at different sizes (*Figure 3B* and *Figure 3—figure supplement 1*). Namely, around 2.5% of seeds completely failed to develop a seed coat and another 20–30% of seeds did not reach full WT size (*Figure 3C–E*). No differences were observed in the integument cell number between the mutant and WT ovules (*Figure 1—figure supplement 2*), indicating that cell expansion following fertilization is affected in the *wei/tar* mutant. These observations strongly support the idea that post-fertilization production of auxin in the fertilization products is necessary for the early stages of seed coat development in *Arabidopsis*, but that auxin signalling does not seem to be required for full seed expansion until later developmental stages.

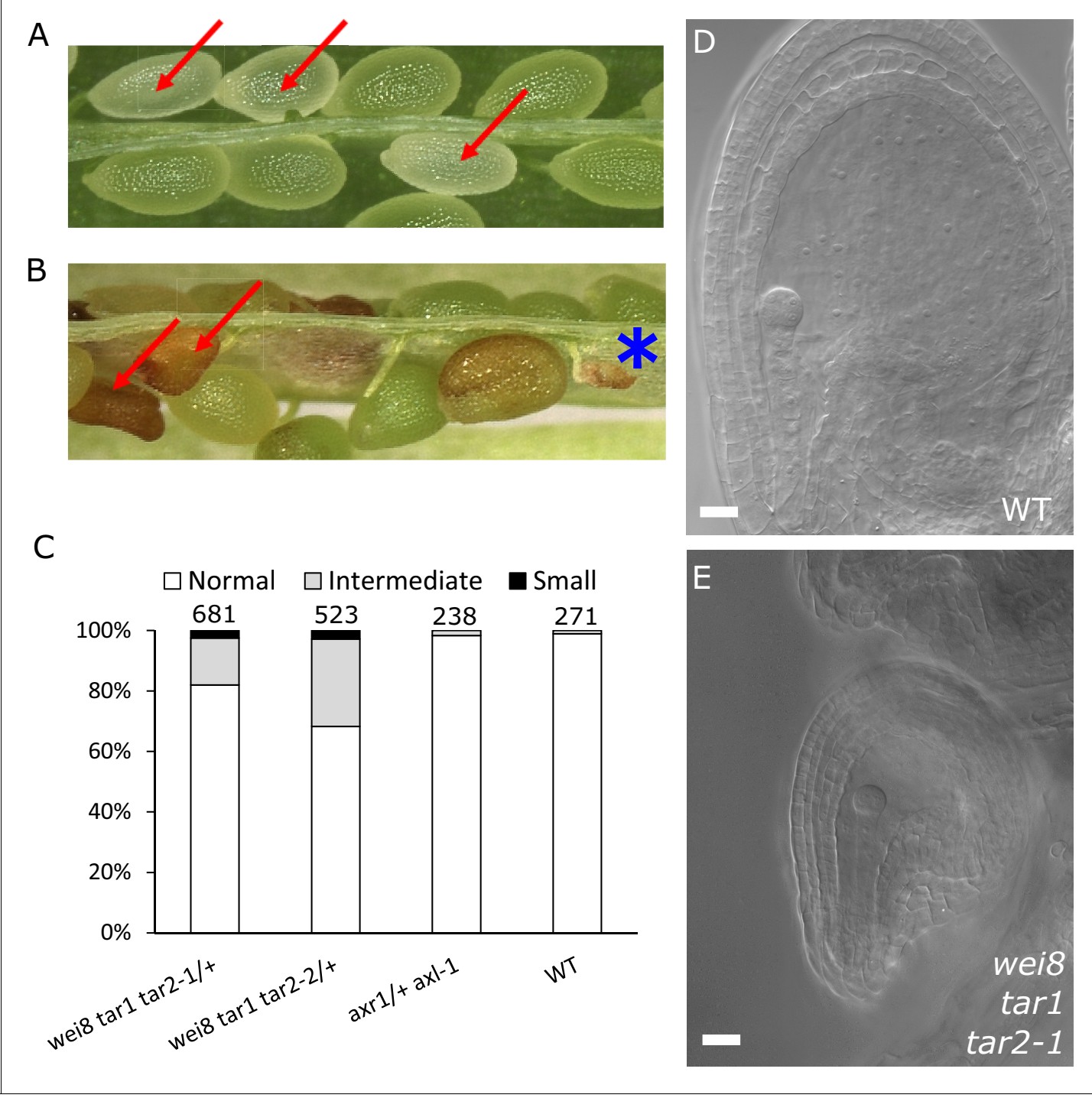

**Figure 3.** Auxin is necessary for seed coat development. (A) Opened silique of an *axr1 axl-1/+* mutant showing aborting white seeds (red arrows) that undergo full seed coat development. (B) Opened silique of a *wei8 tar1 tar2-1/+* mutant showing seeds that do not develop a seed coat (asterisk) and seeds aborting at an intermediate stage compared to WT (red arrows). (C) Determination of final seed size in auxin biosynthesis and signaling mutants and WT. Numbers on top indicate total seeds counted. (D) WT seed at 2 DAP, showing a developed seed coat. (E) *wei8 tar1 tar2-1* seed at 2 DAP that failed to develop seed coat. Bars indicate 20 μm.

The following figure supplement is available for figure 3:

**Figure supplement 1.** Seed size is affected in auxin mutants.

Previous work had revealed that the endosperm, rather than the embryo, is solely responsible for triggering the seed coat development in *Arabidopsis* (*Weijers et al., 2003*; *Ingouff et al., 2006*; *Roszak and Köhler, 2011*). Thus, our data showing that auxin biosynthesis after fertilization is necessary for seed coat development, strongly supports our hypothesis that auxin needs to be produced in the endosperm to initiate seed coat development. Additionally, we previously showed that seeds mutant for *axr1 axl-1* have endosperm proliferation defects (*Figueiredo et al., 2015*). Our observations reported here that these seeds develop a seed coat similar to that of WT seeds indicates that the proliferation of the endosperm per se is not required for seed coat initiation. This is in line with previous observations that mutants with severe endosperm proliferation defects, such as the *titan* mutants, still develop a seed coat (*Liu and Meinke, 1998*).

## Production of auxin in the central cell drives seed coat development

To test whether production of auxin in the central cell is sufficient to trigger seed coat development, we expressed the auxin biosynthesis genes *TAA1* and *YUC6* under control of the central cell and early endosperm specific promoter *DD65* (*Steffen et al., 2007*; *Figueiredo et al., 2015*) and investigated the autonomous seed coat development at 5 DAE. Indeed, *DD65::TAA1; DD65::YUC6* expressing ovules initiated seed coat development, as evidenced by a significant increase in size (*Figure 4A–B*) and the production of protoanthocyanidins (*Figure 4C–D*). The integument cell number was not affected in the transgenic lines (*Figure 1—figure supplement 2*). The increased ovule size in *DD65::TAA1; DD65::YUC6* expressing lines correlated with an increased auxin signaling in sporophytic tissues, revealed by the activation of *DR5v2* in the integuments as well as the removal of the DII:VENUS signal in R2D2 expressing lines (*Figure 4—figure supplement 1*). Ovules expressing *DD65::TAA1; DD65::YUC6* initiate central cell division in the absence of fertilization (*Figueiredo et al., 2015*). However, the autonomous central cell division in these ovules only occurs at 6 DAE, while at 5 DAE seed coat development did clearly initiate. This data reveal that seed coat development precedes replication of the central cell, further indicating that endosperm development per se is not required for the initiation of seed coat development.

Our observations demonstrate that ectopic production of auxin in the central cell is sufficient to drive seed coat development without fertilization. Additionally, the transgenic lines showed pronounced parthenocarpic growth of the gynoecium (*Figure 4E*), supporting the hypothesis that endosperm-produced auxin likely acts as a signal to the maternal tissues, inducing fruit development (*Dorcey et al., 2009*).

## *agl62* seeds show retention of auxin in the endosperm

Our data strongly suggest that production of auxin in the developing endosperm is likely to be the trigger for seed coat development. To further challenge this hypothesis, we investigated plants mutant for the MADS-box transcription factor AGL62. *agl62* seeds abort early after fertilization (around 3–4 DAP), correlating with early endosperm cellularization (*Kang et al., 2008*) and failure to develop a seed coat (*Roszak and Köhler, 2011*). The *agl62* mutant did not show defects in integument cell number, implying compromised cell expansion following fertilization (*Figure 1—figure supplement 2*). To test whether the early cellularization of *agl62* was the cause for the defects in seed coat development, we crossed plants mutant for *agl62/+* with a mutant for *HALLIMASCH* (*hal/+*), whose seeds fail to undergo endosperm cellularization (*Mayer et al., 1999*). In *agl62* seeds the endosperm cellularizes at around 3 DAP, but in double mutant *agl62 hal* seeds this early cellularization phenotype is reverted (*Figure 5A–C*). However, similarly to the single *agl62* mutant, *agl62 hal* seeds do not develop a seed coat and remain small. We thus propose that early cellularization of *agl62* endosperm is a consequence, rather than a cause of the failure to develop seed coat. Given that AGL62 is solely expressed in the endosperm (*Kang et al., 2008*; *Roszak and Köhler, 2011*), we hypothesized whether this transcription factor could be involved in regulating the pathways driving seed coat development. Consistently, GA signaling was not activated in *agl62* integuments, as evidenced by the maintained GFP signal of the RGA DELLA reporter (*RGA::GFP:RGA*) in *agl62* seeds (*Figures 1G* and *5D*). Importantly, also auxin signaling was impaired in *agl62* integuments, which failed to express the *DR5::VENUS reporter*. Strikingly however, while no VENUS signal was observed in *agl62* integuments, we could clearly observe VENUS expression in the endosperm of *agl62* seeds (*Figure 5E–F*), contrasting the absence of VENUS activity in WT seeds. These results strongly

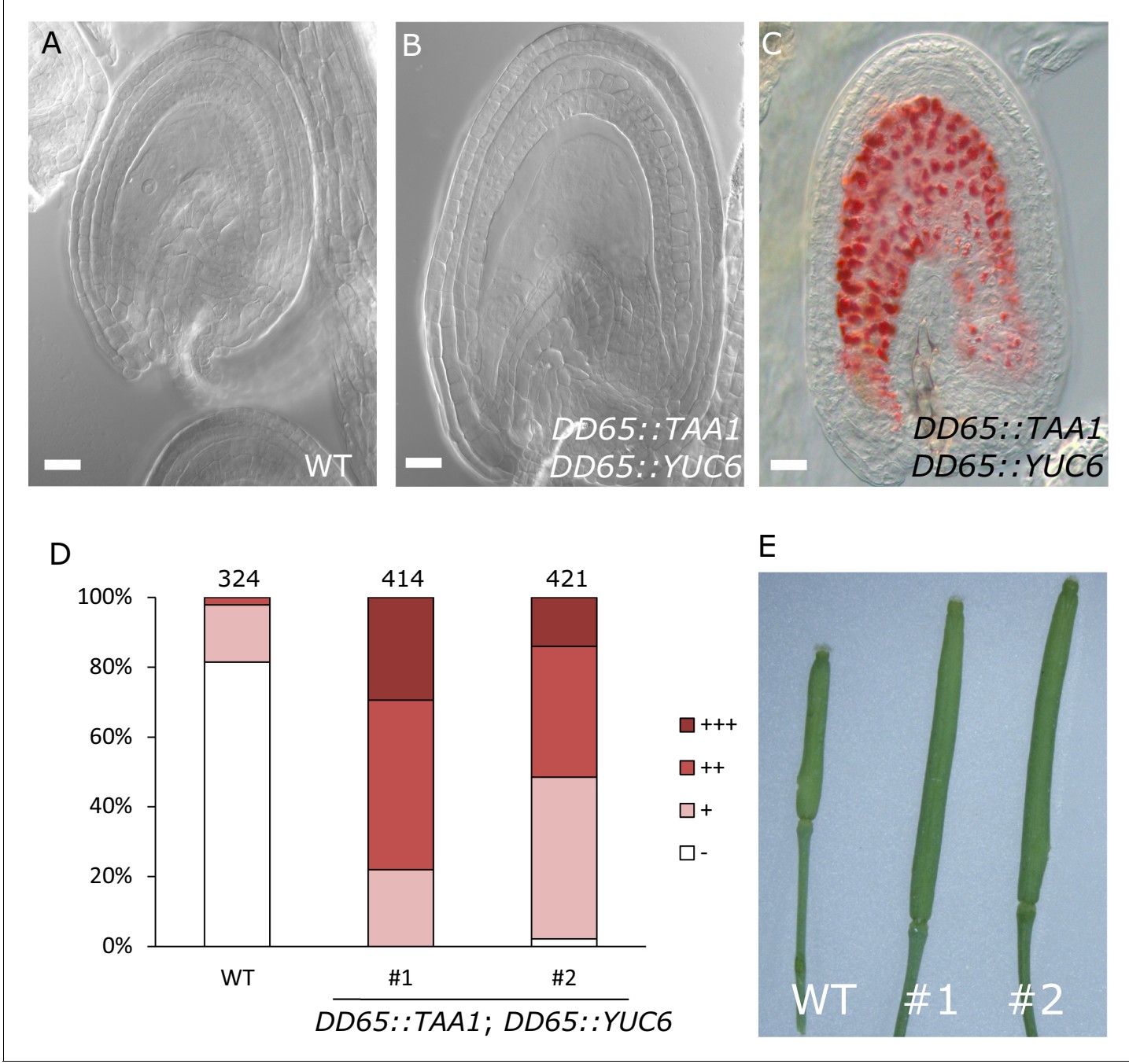

**Figure 4.** Auxin production in the central cell triggers seed coat development. (A–B) Ovule development at 5 DAE: (A) WT and (B) *DD65::TAA1; DD65::YUC6*. (C) Vanillin-stained ovules of *DD65::TAA1; DD65::YUC6* at 5 DAE. Bars indicate 20 µm. (D) Scoring of vanillin staining in WT and two independent transgenic lines expressing *DD65::TAA1; DD65::YUC6*. The scoring was done as indicated in *Figure 2B*. Numbers on top indicate total ovules counted. (E) Pistil size in WT and the two transgenic lines expressing *DD65::TAA1; DD65::YUC6* at 5 DAE.

The following figure supplement is available for figure 4:

**Figure supplement 1.** Auxin reporters in the *DD65::TAA1 DD65::YUC6* transgenic lines.

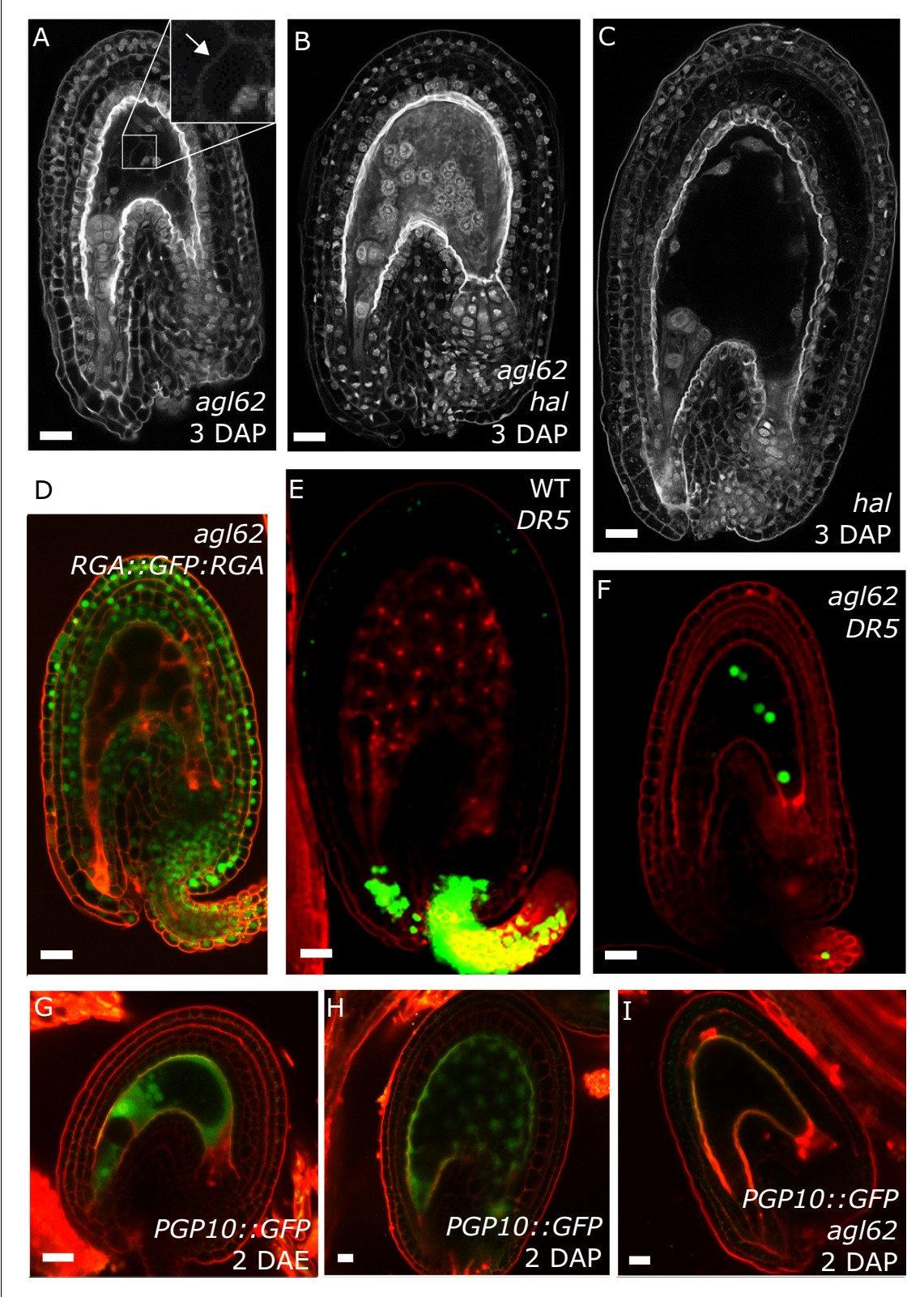

**Figure 5.** *agl62* seeds fail to develop a seed coat. (**A–C**) Endosperm cellularization as determined by Feulgen staining in *agl62* (**A**), *agl62 hal* (**B**) and *hal* (**C**) seeds at 3 DAP. Inlay in (**A**) shows cellularized endosperm (arrow indicates cell wall). (**D**) Expression of *RGA::GFP:RGA* in *agl62* mutant seeds at 1 DAP. See *Figure 1* for expression of the same reporter in WT seeds. (**E–F**) *DR5::VENUS* expression in WT (**E**) and *agl62* (**F**) seeds at 3 DAP. (**G–I**) *PGP10::GFP* expression in ovules (**G**) and in WT (**H**) and *agl62* (**I**) seeds at 2 DAP. Bars indicate 20 μm. Red staining is PI.

*Figure 5 continued on next page*

*Figure 5 continued*

The following figure supplement is available for figure 5:

**Figure supplement 1.** Seed abortion in *agl62* expressing *AGL62::PGP10*.

suggest that in WT seeds, auxin is produced in the endosperm after fertilization and quickly exported into the integuments. However, in the absence of AGL62, auxin export is impaired and it accumulates in the endosperm at sufficiently high levels to activate *DR5* expression.

Our observation that auxin is trapped in the endosperm of *agl62* seeds suggests that AGL62 regulates the expression or activity of auxin transporters in the endosperm. We therefore investigated the transcriptome of *agl62* seeds at 30 hr after manual pollination and searched for genes encoding putative auxin transporters that were downregulated in this mutant compared to WT seeds. We found that *PGP10*, a gene coding for an ABCB-type transporter was significantly downregulated in *agl62* seeds (log2FC = −3.27, p-value=−2.83E-41; See also *Table 1—source data 1*). PGP-type transporters are active auxin transporters (*Geisler et al., 2005*; *Lin and Wang, 2005*), raising the possibility that lack of *PGP10* expression could account for the accumulation of auxin in the endosperm of *agl62* seeds.

In order to test whether *PGP10* was expressed in the endosperm, we developed a *PGP10::GFP* reporter and investigated its expression in ovules and seeds of *Arabidopsis*. Consistently, *PGP10* was expressed before fertilization in the central cell and its expression was maintained in the endosperm after fertilization (*Figure 5G–H*). The expression of *PGP10* required AGL62 function, as *PGP10* expression was not detectable in *agl62* seeds at 2 DAP (*Figure 5I*), in agreement with the *agl62* transcriptome data. Nevertheless, the *PGP10* expression in *agl62* under control of the *AGL62* promoter was not sufficient to induce seed coat development (*Figure 5—figure supplement 1*), suggesting that additional factors are required to activate PGP10 function, similarly as previously reported for other PGPs (*Bouchard et al., 2006*; *Wu et al., 2010*; *Wang et al., 2013*).

## Auxin and sporophytic PRC2s work in the same pathway during seed coat development

Previous observations revealed that sporophytic PRC2s exert a block on seed coat development that is lifted following fertilization (*Roszak and Köhler, 2011*). Lack of the core PRC2 subunits VRN2 and EMF2 results in a dosage-dependent autonomous seed coat development, as visualized by staining of protoanthocyanidins using vanillin (*Figure 6A–C*). Thus, lack of PRC2 function and ectopic auxin are sufficient to initiate seed coat development in the absence of fertilization; we therefore asked whether auxin and PRC2 act in the same pathway initiating seed coat development. To test this hypothesis, we generated triple mutant plants that have reduced PRC2 function and produce auxin ectopically in the integuments by combining mutant alleles for *yuc6-2d, vrn2/-* and *emf2/+*. No differences in integument cell number were observed in these mutants, compared to WT, implying that autonomous ovule growth is mediated by cell expansion (*Figure 1—figure supplement 2*). By scoring protoanthocyanidin production as a proxy for seed coat formation we compared the number of ovules forming autonomous seed coat in this triple mutant to those formed in the *yuc6-2d* single mutant and the *vrn2/- emf2/+* double mutant. Ovules of the triple mutant and the *yuc6-2d* single mutant initiated autonomous seed coat development at similar frequencies (*Figure 6D*), suggesting that auxin and PRC2 act in the same pathway.

Based on the results showing that auxin is the endosperm-derived seed coat initiation signal, it seems most likely that auxin acts upstream of PRC2 during seed coat development. That being the case, autonomous seed coat development in mutants for the PRC2 core components VRN2 and EMF2 should not coincide with the activation of auxin signaling in the integuments. To test this hypothesis, we crossed the *DR5v2* and R2D2 auxin reporters into the *vrn2 emf2* double mutant background. As predicted, despite that *vrn2/+ emf2/+* ovules initiated seed coat development at 5 DAE, there was no detectable expression of *DR5v2*, contrasting to its expression post-fertilization (*Figure 6E–F* and *Figure 1B–C*). Similarly, we did not observe an efficient removal of the DII:VENUS signal from *vrn2/- emf2/+* ovules, unlike what happens post-fertilization (*Figure 6G* and *Figure 1E*).

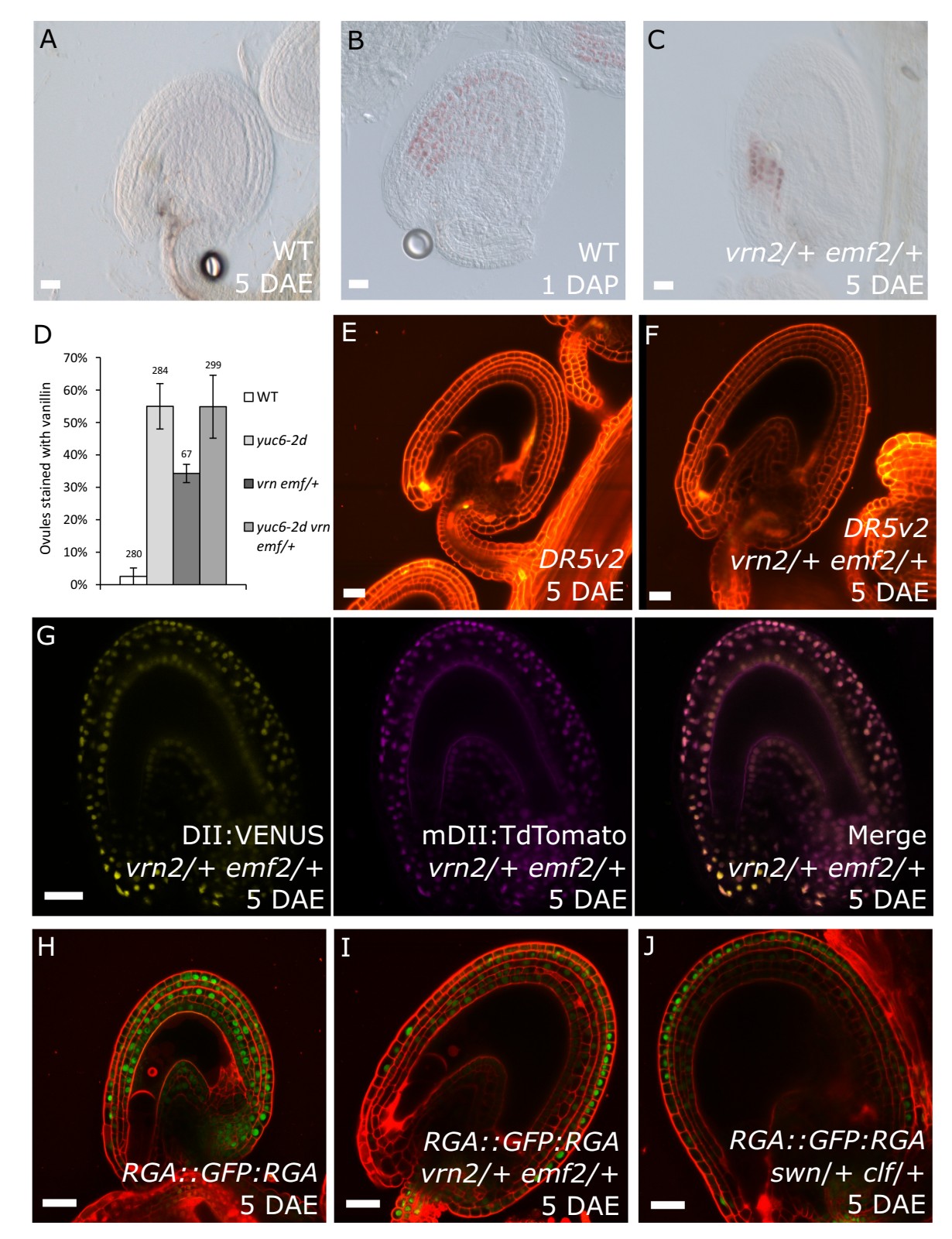

**Figure 6.** PRC2 and auxin work in the same pathway during seed coat development. (A–C) Vanillin-stained WT ovule at 5 DAE (A), WT seed at 1 DAP (B) and *vrn2/+ emf2/+* autonomous seed at 5 DAE (C). (D) Percentage of vanillin-stained ovules per silique at 6 DAE in WT and mutant lines. Bars indicate standard deviation. (E–F) *DR5v2::VENUS* activity in WT (E) and *vrn2/+ emf2/+* (F) unfertilized ovules at 5 DAE. Red is PI. See also *Figure 1* for *DR5v2::VENUS* activity in fertilized seeds. (G) R2D2 reporter activity in autonomous seeds of PRC2 mutant *vrn2/+ emf2/+* at 5 DAE. See also *Figure 1*.
*Figure 6 continued on next page*

Figure 6 continued

(H–J) *RGA::GFP:RGA* activity at 5 DAE in WT ovules (H) or autonomous seeds of mutants for sporophytic PRC2 components: *vrn2/+ emf2/+* (I) and *swn/+ clf/+* (J). Red staining is PI. Bars indicate 20 μm.

The following figure supplements are available for figure 6:

**Figure supplement 1.** Relative expression of genes involved in GA biosynthesis (GA20ox2), GA catabolism (GA2ox2 and GA2ox6) and GA signaling (RGA) between vrn2 emf2/+ and WT ovules.

**Figure supplement 2.** Expression of the RGA reporter in unfertilized ovules.

These observations strongly support the hypothesis that auxin acts upstream of the PRC2 complex, and might have a role in removing the PRC2-block on seed coat development. To investigate how PRC2 function is depleted following either fertilization or application of auxin, we analyzed transgenic lines expressing translational reporters for the PRC2 components MSI1 and SWN, as well as for the PRC2-associated protein LHP1 (*Derkacheva et al., 2013*). Before fertilization there was a strong expression of all three reporters in the integuments, which markedly decreased following fertilization (*Figure 7*). Interestingly, a similar decrease in reporter activity was observed in unfertilized ovules following exogenous application of 2,4-D (*Figure 7C,F and I*), but no changes were observed in mock-treated ovules (*Figure 7—figure supplement 1*). These observations support the view that post-fertilization auxin transport to the integuments leads to the removal of PRC2 and LHP1 proteins. To test whether the downregulation of PRC2 components occurred at the transcriptional level, we tested the expression of genes coding for sporophytic PRC2 components. All the genes tested were strongly downregulated after fertilization (*Figure 7J*) as well as after exogenous application of 2,4-D in unfertilized ovules (*Figure 7—figure supplement 1*). This data reveals that the auxin-dependent removal of PRC2 function is likely to occur at the transcriptional level; nevertheless, an additional post-translational regulation of PRC2 components by auxin cannot be completely ruled out.

Given that GA signalling is downstream of auxin during seed development (*Figure 1F–H*), we asked whether the seed coat growth in mutants for sporophytic PRC2s would coincide with the ectopic activation of GA signaling in the integuments, similarly to what happens following application of auxin. Indeed, the GA biosynthesis gene *GA20ox2* was upregulated in *vrn2/- emf2/+* ovules while the GA catabolism genes *GA2ox6* and *GA2ox2* were downregulated when compared to WT (*Table 1—source data 1* and *Figure 6—figure supplement 1*). To further test this hypothesis, we crossed the GA reporter *RGA::GFP:RGA* into mutants deficient for either PRC2 components VRN2 and EMF2 or SWN and CLF, and analyzed *GFP* expression in unfertilized ovules. Indeed, when compared to WT ovules, the ovules mutant for *vrn2/+ emf2/+* or *swn/+ clf/+* had a strongly reduced GFP:RGA signal, particularly in the innermost layers of the integuments (*Figure 6H–J*) This was observed both at 5 DAE, when the autonomous seeds were significantly larger than WT ovules, but also at 2 DAE, indicating that activation of GA signalling takes place shortly after anthesis (*Figure 6—figure supplement 2*). This decreased signal is most likely due to RGA protein degradation, since the *RGA* gene was not downregulated in the *vrn2/- emf2/+* mutant (*Table 1—source data 1* and *Figure 6—figure supplement 1*). We conclude that GA signaling is activated in the absence of fertilization, when PRC2 function is removed from the integuments.

## Discussion

In this manuscript we reveal that auxin is the missing link connecting fertilization and seed coat development. As the maternal integuments do not take part in the fertilization process, the signal that coordinates the development of the fertilization products with the maternal integuments remained elusive (*Figueiredo and Köhler, 2016*). Based on previous research it seemed likely that the signal is generated in the endosperm, as endosperm development is required to drive seed coat initiation in *Arabidopsis* (*Weijers et al., 2003*; *Ingouff et al., 2006*; *Roszak and Köhler, 2011*). It was furthermore known that the presence of the paternal genome is necessary for the seed coat to develop, as mutants for *cdka;1* and *fbl17*, in which one of the sperm cells fails to undergo karyogamy with the central cell, do not develop a seed coat (*Gusti et al., 2009*; *Aw et al., 2010*). These

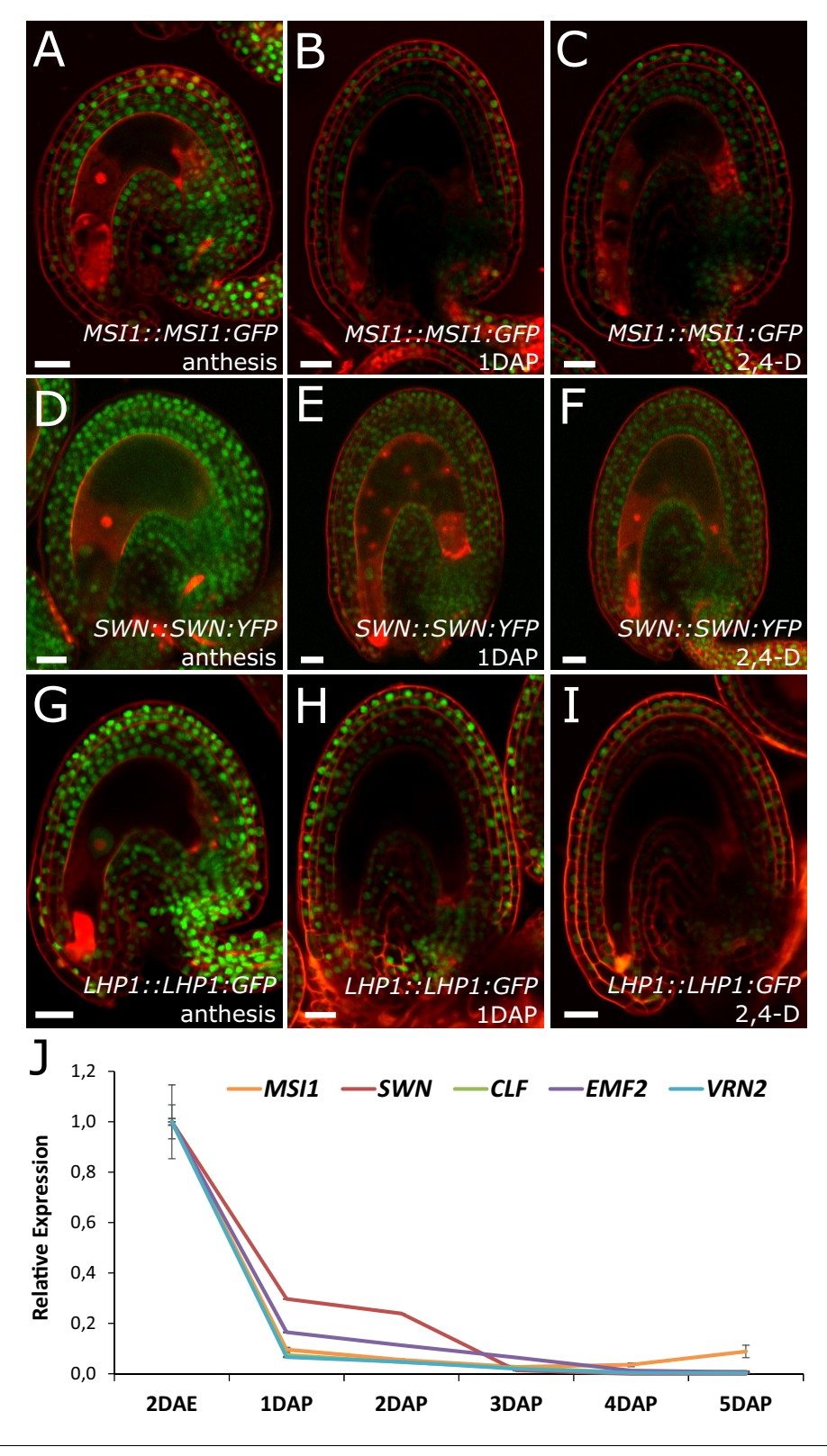

**Figure 7.** PcG activity is reduced in the developing seed coat. (A–I) Expression of *MSI1:MSI1::GFP* (A–C), *SWN:SWN::GFP* (D–F) and *LHP1:LHP1::GFP* (G–I) reporters in 2 DAE ovules (A,D and G), 1 DAP seeds (B,E and G) and 2,4-D treated ovules at 3 DAE (C, F and I). For mock controls see figure supplement 1. Red staining is PI. Bars indicate 20 µm. (J) Relative expression of sporophytic *PRC2* genes during seed development, as determined by
*Figure 7 continued on next page*

*Figure 7 continued*

RT-qPCR. Gene expression was measured at 2DAE and from 1 to 5 DAP and was normalized to the 2 DAE time-point. Error bars indicate standard deviation.

The following figure supplement is available for figure 7:

**Figure supplement 1.** Expression of PcG reporters in unfertilized ovules.

observations strongly suggest that the signal driving seed coat development should be dependent on genes that are paternally-expressed in the endosperm. Auxin biosynthesis following fertilization of the central cell is dependent on the paternally-expressed genes *YUC10* and *TAR1* (*Figueiredo et al., 2015*), raising the hypothesis that auxin could be the trigger for seed coat initiation. In this manuscript we provide multiple lines of evidence supporting this hypothesis: (i) auxin rapidly accumulates in the integuments after fertilization, (ii) impaired auxin biosynthesis but not auxin signaling in the endosperm causes defects in seed coat development, (iii) ovules of transgenic lines producing auxin ectopically in the central cell initiate seed coat development without fertilization, (iv) failure of seed coat development in the *agl62* mutant correlates with failure to export auxin. Collectively, our data strongly support the hypothesis that auxin generated in the fertilized central cell/endosperm is the trigger for seed coat formation. Impaired auxin signaling in the endosperm did not impair seed coat formation, strongly supporting the hypothesis that auxin is exported to the integuments, where it initiates downstream signaling events. We could show that the ABCB transporter *PGP10* is expressed in the fertilized central cell and its expression depends on the type I MADS-box transcription factor AGL62, suggesting that PGP10 may regulate the export of auxin from the endosperm to the integuments. PGPs are known to depend on additional proteins for their localization and function (*Bouchard et al., 2006*; *Wu et al., 2010*; *Wang et al., 2013*), likely explaining our failure to restore seed coat development in *agl62* by ectopically expressing *PGP10*. It is furthermore possible that additional AGL62 targets are required for successful auxin export from the endosperm and are therefore lacking in the *agl62* mutant. Future research will focus on identifying the remaining factors required for PGP10 function.

Post-fertilization auxin production was shown to lead to activation of GA biosynthesis in the ovules, which is then transported to the valves to promote silique growth (*Dorcey et al., 2009*). Our observation that auxin produced in the central cell is sufficient to drive parthenocarpic growth of the silique couples the fertilization event to fruit development.

We have previously shown that auxin drives endosperm development in *Arabidopsis* (*Figueiredo et al., 2015*). We now propose a dual role for post-fertilization auxin production, both in driving endosperm and seed coat development. Additionally, the *agl62* mutant has defects both in seed coat development in fertilized ovules and endosperm development in autonomous seeds. This suggests that AGL62, like auxin, is involved in mediating both endosperm and seed coat developmental pathways. It was recently proposed that AGL62 in the endosperm also mediates non-autonomous cell signalling driving nucellus degeneration (*Xu et al., 2016*). This transcription factor thus seems to be a central player in modulating diverse developmental pathways during seed development.

While ectopic auxin production in the central cell was sufficient to initiate seed coat development, the autonomous seeds did not reach the final size of a fertilized seed and collapsed at around 6–7 DAE. This implies that auxin is sufficient to initiate seed coat development, but that additional signals are necessary to support its full expansion and prevent the degeneration of the ovule. Recent work has shown that the innermost layer of the outer ovule integument perceives mechanical cues from the expanding endosperm, which affects cell wall thickening and GA metabolism in that cell layer (*Creff et al., 2015*). It is therefore an attractive hypothesis that auxin is the initial paternally-derived signal that drives the integument-to-seed coat developmental transition, and that mechanical pressure from the syncytial endosperm sustains seed coat growth. This view is supported by observations showing that mutants affected in syncytial endosperm proliferation have smaller seeds when compared to WT plants (*Garcia et al., 2003*; *Luo et al., 2005*; *Zhou et al., 2009*; *Wang et al., 2010*). Conversely, a gain-of-function mutation in *SHORT HYPOCOTYL UNDER BLUE1* results in increased proliferation of the syncytial endosperm delayed cellularization and increased seed size

(*Zhou et al., 2009*). Additionally, it cannot be ruled out that other signalling pathways contribute to sustain seed coat growth, such as small peptide-mediated signalling (*Ingram and Gutierrez-Marcos, 2015*; *Figueiredo and Köhler, 2016*)

Mutants in sporophytic PRC2 components develop seed coat autonomously, implying that the PRC2 imposes a block on seed coat development that has to be lifted upon fertilization (*Roszak and Köhler, 2011*). Here, we demonstrate that ectopic auxin can bypass the PRC2 block on seed coat development, which leads to activation of GA signaling and production of protoanthocyanidins in the integuments. We propose that transport of auxin from the developing endosperm is required for the removal of PRC2 function in the integuments, allowing for the development of the seed coat by activation of GA signaling (*Figure 8*). Also in mammals PRC2 regulates the transition from proliferation to differentiation during organogenesis and PRC2 function has to be removed to allow differentiation of embryonic stem cells (*Leeb et al., 2010*; *Aldiri and Vetter, 2012*). We further show that exogenous auxin is sufficient to remove PRC2 function from the ovule integuments. Our gene expression analysis revealed that sporophytic PRC2-coding genes are downregulated after fertilization, likely due to increased auxin signalling in the integuments. Auxin is known to extensively modulate transcription (*Weijers and Wagner, 2016*), but the mechanism by which it regulates PRC2-coding genes is yet to be understood. In fact, although there are reports on the transcriptional regulation of PRC2 genes in animals (*Neri et al., 2012*), there is very little information on how the plant PRC2 activity is regulated. Thus, we provide the first evidence on the modulation of the PRC2 activity by a plant hormone, as a way to modulate developmental transitions.

In conclusion, we have discovered that auxin produced in the fertilized central cell is required to induce seed coat formation after fertilization and most likely acts as mobile signal linking fertilization with the differentiation of the surrounding sporophytic tissues.

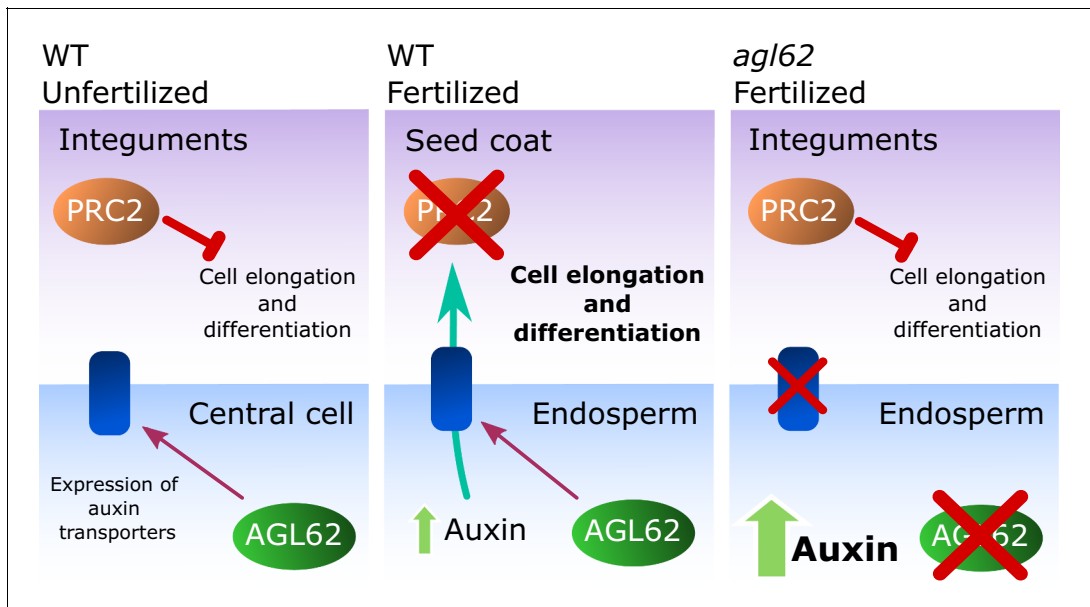

**Figure 8.** Model for the post-fertilization mechanism driving seed coat development. Before fertilization (left panel) sporophytic PRC2 represses seed coat developmental pathways. The fertilization of the central cell (middle panel) results in the production of auxin that is exported to the integuments, lifting the PRC2 block on seed coat development and allowing for cell elongation and differentiation. However, this process is highly dependent on the activity of the transcription factor AGL62, since *agl62* seeds fail to export auxin from the endosperm. This leads to auxin accumulation and consequently, failure to develop a seed coat (right panel). We propose that AGL62 activity is important for the expression, and eventually for the activity, of endosperm-specific auxin-exporters.

## Materials and methods

### Plant material, growth conditions and treatments

The *Arabidopsis thaliana* mutants and reporters used were described previously: *emf2*-5 (*Yang et al., 1995*), *vrn2-1* (*Chandler et al., 1996*), *swn-3* (*Chanvivattana et al., 2004*), *clf-9* (*Goodrich et al., 1997*), *wei8-1/- tar1/- tar2-1/+* and *wei8-1/- tar1/- tar2-2/+* (*Stepanova et al., 2008*), *axr1-12/+ axl-1/-* (*Dharmasiri et al., 2007*), *agl62-2/+* (*Kang et al., 2008*), *hal/+* (*Mayer et al., 1999*), *SWN::SWN:YFP* (*Wang et al., 2006*), and *LHP1::LHP1:GFP* (*Kotake et al., 2003*).

Seeds were sterilized in 5% commercial bleach and 0.01% Tween-20 for 10 min and washed three times in sterile ddH$_2$O. Sterile seeds were plated on ½ MS-medium (0.43% MS-salts, 0.8% Bacto Agar, 0.19% MES hydrate and 1% Sucrose; when necessary, the medium was supplemented with the appropriate antibiotics) and stratified at 4°C in the dark for 48 hr. Plates were then transferred to a growth chamber (16 hr light/8 hr dark; 110 µmol.s$^{-1}$.m$^{-2}$; 21°C; 70% humidity). After 10 days seedlings were transferred to soil and grown in a growth chamber (16 hr light/8 hr dark; 110 µmol.s$^{-1}$.m$^{-2}$; 21°C; 70% humidity).

The solutions used for hormone treatments contained 5% EtOH, 0.05% Tween-20 and 200 µM of 2,4-Dichlorophenoxyacetic acid (2,4-D) or 200 µM Gibberellic acid (GA$_3$). In all experiments a mock control was run in parallel. Flowers were emasculated two days before anthesis and treated at 2 DAE by covering the whole pistils with 2 µL of the respective solution. The treated pistils were then collected at the specified time-points and processed for microscopy analyses, as described below.

### Transcriptome analysis

WT (L*er*) and double mutant *vrn2/- emf2/+* plants were emasculated at 1–2 days prior to anthesis. Two days after emasculation half of the WT plants were hand-pollinated. Unfertilized ovules and young seeds were dissected from the siliques at 4 DAE and 2 DAP, respectively, collected in RNA-later (Sigma-Aldrich Chemie GmbH, Switzerland) and ground frozen in a Silamat S5 for 3 times 7 s. Three biological replicates were generated per sample. RNA was extracted using the QIAGEN RNeasy KIT followed by on-column DNase treatment (QIAGEN Instruments AB, Switzerland). Labeling and hybridization to AGRONOMICS1 arrays (Affymetrix UK Ltd., United Kingdom) was done as described (*Hennig et al., 2003*). Signal values were derived from Affymetrix CEL files using RMA (RRID:SCR_008549) (*Rehrauer et al., 2010*). All data processing was done using the statistics package R (version 2.6.2, RRID:SCR_001905) that is freely available at http://www.rproject.org/. Quality control was done using the affyQCReport package in R (RRID:SCR_001318). In addition, coefficients of variation (cv) were calculated between replicates as a quantitative measure of data quality and consistency between replicates as described previously (*Köhler et al., 2003*). Differentially expressed genes were identified using the limma package in R (RRID:SCR_010943) (*Smyth, 2004*). Multiple-testing correction was done using the q-value method (*Storey and Tibshirani, 2003*). Probesets were called significantly differentially expressed when q < 0.05. The genes used in the analysis were selected by overlapping the list of significantly upregulated genes in 2 DAP seeds and in *vrn2 emf2* ovules. The list of commonly upregulated genes was then used to determine enriched GO-terms using AtCOECIS. Only significantly enriched biological processes were considered (p-value<0.05). REVIGO was used to remove redundant GO terms and summarize the list. For the transcriptome analysis of *agl62* seeds, we made use of homozygous *agl62* mutant plants that arise at low frequency and that form small amounts of viable *agl62/-* seeds. *agl62* plants were emasculated 1–2 days prior to anthesis, hand-pollinated and seeds were harvested 30 hr after pollination. Three biological replicates were generated per sample. Isolation of RNA and preparation of libraries was done as previously described (*Wolff et al., 2015*). RNA was sequenced at the Functional Genomics Center Zurich (Switzerland) on an Illumina HiSeq2000 (Ilumina, San Diego, USA) on two lanes in 100 bp paired end fashion. Sequencing reads were aligned to the TAIR 10.0 version of the Arabidopsis reference genome (Col-0) using TopHat (RRID:SCR_013035) (*Trapnell et al., 2009*). Only uniquely mapping reads were used for further analysis. Differentially expressed genes were identified using the DESeq package (*Anders and Huber, 2010*). Analysis of GO categories was performed using AtCOECiS (*Vandepoele et al., 2009*) and REVIGO (RRID:SCR_005825) (*Supek et al., 2011*). To enrich for biologically relevant changes, probesets with signal log ratio (SLR) > 0.6 were selected for

the transcriptome analysis of *vrn2/- emf2/+* and with SLR<−1 for the analysis of *agl62*. Affymetrix data and sequencing reads are deposited as CEL and fastq files, respectively, in the Gene Expression Omnibus (RRID:SCR_005012) (datasets GSE85751 and GSE85848).

## Cloning and generation of transgenic plants

To clone the promoters of *PGP10*, *BAN* and *AGL62*, as well as the coding region of *GA3ox1* and *PGP10*, WT Col-0 genomic DNA was used as a template. The primer sequences can be found in *Table 2*. The amplified fragments were purified from the gel, recombined into the donor vector (pDONR221) to create entry clones, and sequenced. Gateway cloning was done according to the manufacturer's instructions (Invitrogen, Fisher Scientific, Sweden). The promoter of *PGP10* was recombined into vector pB7FWG.0. The coding region of *GA3ox1* and *PGP10* were recombined into vector pB7WG2, where the 35 s promoter was replaced by that of *BAN* and *AGL62*, respectively, using the restriction enzymes *Sac*I and *Spe*I. The MSI1 translational reporter was obtained by replacing the TAP fragment in the *MSI::MSI1:TAP* construct with GFP (*Bouveret et al., 2006*). The *BAN:: GFP* reporter was generated by recombining the amplified promoter sequence of *BAN* and the coding sequence of *GFP* into the pBGW vector. Recombination was done using the In-Fusion enzyme (Clontech, Takara Bio Europe, France), following the manufacturer's instructions.

The constructs were transformed into *Agrobacterium tumefaciens* strain GV3101 and *Arabidopsis* plants were transformed using the floral dip method (*Clough and Bent, 1998*). Transformants were selected with the appropriate antibiotics.

---

**Table 2.** Primer sequences.

| Used for | Gene | | Sequence 5′ - > 3′ * |
|---|---|---|---|
| RT-qPCR | AT5G58230 (*MSI1*) | fw | GCCCAAGTTCAGCTTCCTCT |
| | | rv | TTTGTACCTTTCCAGTTGCACA |
| | AT4G02020 (*SWN*) | fw | CAACTCCTCTGGACGAATCAAG |
| | | rv | TCTGTTTTCCAAACCCTCGAGTC |
| | AT4G16845 (*VRN2*) | fw | TCATTCTCACAGAGTCCAGCC |
| | | rv | AGTCATCAAGCATCTGGCGAT |
| | AT5G51230 (*EMF2*) | fw | CGCACTTGATTTGGTGCTGG |
| | | rv | TGTTCATGGTTCGGGCATCA |
| | AT2G23380 (*CLF*) | fw | AAGTACTGCGGTTGCCCAAA |
| | | rv | ACATTCCCGATCTGCAGCAA |
| Cloning | AT1G15550 (*GA3ox1*) | fw | GGGGACAAGTTTGTACAAAAAAGCAGGCTGCAAGATGCCTGCTATGTTA |
| | | rv | GGGGACCACTTTGTACAAGAAAGCTGGGTATCTAATCATTCTTCTCTGTGATTTCT |
| | AT1G61720 (prom*BAN*) Gateway | fw | GGGGACAAGTTTGTACAAAAAAGCAGGCTGAGCTCTAACAGAACCTTACTGTAACACT |
| | | rv | GGGGACCACTTTGTACAAGAAAGCTGGGTACTAGTGAGTCTGGTCCATGGTTGTA |
| | AT1G61720 (prom*BAN*) In-Fusion | fw | CCATGGCCGCGGGATATCAGATTCTTAGGTGAAGACAAG |
| | | rv | CGCTGAATGATTCATGATTGTACTTTTGAAATTACAG |
| | GFP | fw | ATGAATCATTCAGCGAAAACC |
| | | rv | CTTCACCTAAGAATCCATCTAGTAACATAGATGACA |
| | AT1G10680 (prom*PGP10*) | fw | GGGGACAAGTTTGTACAAAAAAGCAGGCTGGCGTTGCGTATAATCCGTT |
| | | rv | GGGGACCACTTTGTACAAGAAAGCTGGGTTTTTCACTTTTGGATATGGAGAGA |
| | At5G60440 (prom*AGL26*) | fw | TAAGCAGAGCTCGAATTGCATCTCGGCAATGAC |
| | | rv | TGCTTAACTAGTTTTTAGTGATATTTGAGAAGCT |
| | AT1G10680 (*PGP10*) | fw | GGGGACAAGTTTGTACAAAAAAGCAGGCTATGCAACCGTCAAATGATCCAG |
| | | rv | GGGGACCACTTTGTACAAGAAAGCTGGGTTTAAGGATGATGGCGCTGC |

*Primer adapters are underlined

## Histological and fluorescence analyses

For clearing of ovules and seeds the whole pistils/siliques were fixed with EtOH:acetic acid (9:1), washed for 10 min in 90% EtOH, 10 min in 70% EtOH and cleared over-night in chloralhydrate solution (66.7% chloralhydrate (w/w), 8.3% glycerol (w/w)). The ovules/seeds were observed under differential interference contrast (DIC) optics using a Zeiss Axioplan or Axioscope A1 microscopes (Carl Zeiss AB, Sweden). The vanillin staining was done on 5 DAE ovules in 1% (w/v) vanillin (4-hydroxy-3-methoxybenzaldehyde) in 6 N HCl. The emasculated pistils were incubated in this solution for 30 min and then the ovules were dissected out and mounted on a microscope slide. Images were recorded using a Leica DFC295 camera with a 0.63x optical adapter (Leica Microsystems, Sweden).

For fluorescence analysis seeds were mounted in 7% glucose. Where indicated, 0.1 mg/mL propidium iodide (PI) was used. Samples were analyzed under confocal microscopy on a Zeiss 780 Inverted Axio Observer with a supersensitive GaASp detector with the following settings (in nm; excitation-ex and emission-em): GFP – ex 488, em 499–525; PI – ex 488/514, em 635–719; YFP (VENUS) – ex 514, em 499–552 for DR5v2 and 525–543 for DII; tdTomato – ex 561, em 599–622. Images were acquired, analyzed and exported using Zeiss ZEN software.

For Feulgen staining of seeds, fixation, staining and embedding were performed as described (*Braselton et al., 1996*). Confocal imaging was performed using a Leica SP1-2, excitation wavelengths were set to 488 nm and detection to 535 nm and longer.

Ovule size and cell number measurements (*Figure 1—figure supplement 2*) were performed through confocal imaging of PI stained ovules. Each ovule was imaged at the median longitudinal plane, and its area was subsequently measured using the ImageJ software (RRID:SCR_002285) (*Schneider et al., 2012*). For each condition, the area of 10 ovules was measured. Average ovule area for each condition was normalized to the average area of mock treated ovules. The determination of integument cell number in all lines was done for the outer layer of the inner integument at 2 DAE and 10 ovules were analyzed per line. Ovule and seed area measurements for WT and auxin mutants (*Figure 3—figure supplement 1*) were performed through seed clearing (as described above) and measured using ImageJ software (RRID:SCR_002285) (*Schneider et al., 2012*). An average of 50 ovules/seeds was measured for each line per time-point.

## RT-qPCR analyses

Ovules/seeds of 25 emasculated or hand-pollinated WT siliques were harvested at 2DAE, 1, 2, 3, 4 and 5 DAP in 20 µL of RNAlater solution (Invitrogen) and ground for 2 min using a TissueLyser II (Qiagen AB, Sweden). Total RNA was extracted using the Qiagen RNeasy kit, followed by DNase I treatment (Qiagen). cDNA was synthesized using the RevertAid First Strand cDNA Synthesis Kit (Thermo Scientific, Fisher Scientific, Sweden). Maxima SYBR Green qPCR Master Mix (Thermo Scientific) was used to perform the qPCR in an iQ5 qPCR system (Bio-Rad Laboratories AB, Sweden). The primers used for the RT-qPCR are described in *Table 2*. PP2A was used as the reference gene. Relative quantification of gene expression was performed as described (*Pfaffl, 2001*). Expression levels for each gene were normalized to the expression level at 2 DAE.

## Acknowledgements

We are indebted to Dolf Weijers for providing the R2D2 system and *DR5v2* reporter prior to publication, to Koji Goto for the LHP1 reporter, to Ramin Yadegari for the SWN reporter and to Miguel Blázquez for providing us with the RGA reporter. This research was supported by a European Research Council Starting Independent Researcher grant (to CK), a grant from the Swedish Science Foundation (to CK), a grant from the Olle Engkvist Byggmästare Foundation (to CK), and a grant from the Knut and Alice Wallenberg Foundation (to CK).

## Additional information

### Funding

| Funder | Author |
| --- | --- |
| European Research Council | Claudia Köhler |

| Vetenskapsrådet | Claudia Köhler |
|---|---|
| Knut och Alice Wallenbergs Stiftelse | Claudia Köhler |
| Olle Engkvist Byggmästare Foundation | Claudia Köhler |

The funders had no role in study design, data collection and interpretation, or the decision to submit the work for publication.

## Author contributions

DDF, Conception and design, Acquisition of data, Analysis and interpretation of data, Drafting or revising the article; RAB, PJR, Conception and design, Acquisition of data, Analysis and interpretation of data; LH, Analysis and interpretation of data, Drafting or revising the article; CK, Conception and design, Analysis and interpretation of data, Drafting or revising the article

# Additional files

### Major datasets

The following datasets were generated:

| Author(s) | Year | Dataset title | Dataset URL | Database, license, and accessibility information |
|---|---|---|---|---|
| Roszak PJ, Köhler C | 2016 | Expression data of vrn2/- emf2/+ and wild-type ovules and seeds | https://www.ncbi.nlm. nih.gov/geo/query/acc. cgi?acc=GSE85751 | Publicly available at the NCBI Gene Expression Omnibus (accession no: GSE85751) |
| Roszak PJ , Köhler C | 2016 | Expression data of agl62/- and wild-type seeds at 30 hrs after pollination | https://www.ncbi.nlm. nih.gov/geo/query/acc. cgi?acc=GSE85847 | Publicly available at the NCBI Gene Expression Omnibus (accession no: GSE85847) |

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
