## [Decision Letter]

Thank you for submitting your article "Auxin production in the endosperm drives seed coat development in Arabidopsis" for consideration by *eLife*. Your article has been reviewed by two peer reviewers, and the evaluation has been overseen by a Reviewing Editor and Christian Hardtke as the Senior Editor. The following individuals involved in the review of your submission have agreed to reveal their identity: Lars Ostergaard (Reviewer #1); Gwyneth Ingram (Reviewer #2).

The reviewers have discussed the reviews with one another and the Reviewing Editor has drafted this decision to help you prepare a revised submission.

Your work contributes to the understanding of how fertilization of the ovule induces formation of the seed coat. Earlier work indicated that the signal would be derived from the endosperm. The finding that the signal is auxin and that part of the pathway of seed coat development involves altering expression of genes encoding Polycomb Group proteins is an important contribution. There are however several issues that need to be addressed for the paper to be suitable for the general audience of *eLife*. Therefore, we encourage resubmission of a suitably modified manuscript which will be reviewed again. We believe that the issues below could be addressed in a reasonable time frame.

1) At the mechanistic level the link to GA is not clear. PRC2 should repress GA biosynthesis in the testa, and yet whether this is the case has not really been directly addressed either experimentally or hypothetically in the Discussion. The data shows that auxin is upstream of gibberellin signaling and PRC2 repression of seed coat differentiation. Moreover, the data in Figure 6 seem to suggest that GA signaling occurs without fertilisation in the integuments of the *vrn2/+ emf2/+* mutant (as opposed to auxin signaling). Do the authors therefore propose the following: Auxin produced in the endosperm is transported to the integument where it inhibits PRC2 gene expression, which then in turn induce GA signaling/synthesis? This is not clear from the text, and as noted above needs to be addressed (perhaps a schematic model could clarify in addition to text in the Discussion).

2) In the last paragraph of the subsection “Auxin and sporophytic PRC2s work in the same pathway during seed coat development”, Figure 6: Although the link to auxin in this paper is strong, the link to GA is not well developed and the results shown here (looking at GFP:RGA in *prc2* mutants) are rather weak. For example, is the expression of the pRGA-GFP:RGA transgene really equivalent at the RNA level in the two types of seeds shown? This should be tested. In addition, the *prc2* mutant seeds shown have grown a lot compared to the wild-type by 5DAE, which can affect "quantitative" imaging, particularly in internal layers where the authors see the largest differences. The authors are not really comparing like with like in this experiment. Would it be possible to look earlier, when the seeds are at a more comparable developmental stage?

3) The experiments on the effects of auxin signaling vs. auxin production in the endosperm (Figure 3 and Figure 4) are elegant in their conception, but the interpretation of the data needs to be related to what has previously been published.

First, one cannot claim that seeds with a homozygous *axr1/- axr1/-* zygotic compartment are the same size as wild-type siblings just by showing a photograph (Figure 3). This should be quantified. Indeed, they look smaller (which would be consistent with the endosperm proliferation defects reported in the mutant in Figueiredo et al. 2015).

The argument that the defects in testa development in the auxin biosynthetic mutant backgrounds are due to endosperm-produced auxin is difficult to justify. This mutant will produce less auxin in the endosperm, but as pointed out in Figueiredo et al. 2015, this lack of auxin production causes major endosperm development problems and it is thus impossible to exclude the possibility that a lack of other endosperm-derived signals could underlie the serious defects in this background. A similar problem is associated with the DD65:TAA1; DD65:YUC1 lines. Figueiredo et al. (Figure 3) 2015 shows that these lines lead to the formation of an autonomous endosperm, thus again it is impossible to exclude an additional indirect effect of endosperm development as the reason for seed coat development. Thus, the conclusions from these experiments are perhaps overstated; although these experiments show that auxin production in the endosperm is needed for seed coat development, they do not prove that auxin itself is the signal. I.e., it is important to make any caveats clear and this will add to the value of your paper.

4) The authors argue that additional factors are required for PGP10 function, which is why the AGL62::PGP10 construct fails to complement the *agl62* mutant. Although this is one possible explanation, it must follow from this experiment that these additional factors are also targets of AGL62 and therefore not present in the *agl62* mutant. Perhaps it would be safer to reformulate with a more general statement that these results demonstrate that other targets of AGL62 are involved in facilitating auxin transport to the integuments. Some of these targets may indeed encode the additional factors required for PGP10 function.

5) In the last paragraph of the subsection “Auxin and GA trigger autonomous seed coat development”, it is stated that it is expected that the expression of hormone biosynthesis gene expression is specific to the integuments in BAN::GA3OX1 and not expressed in endosperm in the *yuc6-2d* line. Given the importance of this result, it would be useful if the authors could confirm this for instance by an in situ hybridisation using these lines.

6) In the second paragraph of the subsection “Auxin and sporophytic PRC2s work in the same pathway during seed coat development” and Figure 7: These results are over-interpreted. 1) The fact that transcriptional control of the PRC2-encoding genes is observed doesn't allow the authors to exclude regulation at the protein level. 2) More importantly, the fact that transcription of these genes decreases after pollination (there is a typo in the axis label of Figure 7), does not allow the authors to conclude that auxin-dependent removal of PRC2 function occurs at the level of transcription. The authors need to include either auxin treatments, or relevant mutant/transgenic lines in their Q-PCR experiment to allow them to draw this conclusion.

---

## [Author Response]

*1) At the mechanistic level the link to GA is not clear. PRC2 should repress GA biosynthesis in the testa, and yet whether this is the case has not really been directly addressed either experimentally or hypothetically in the Discussion. The data shows that auxin is upstream of gibberellin signaling and PRC2 repression of seed coat differentiation. Moreover, the data in Figure 6 seem to suggest that GA signaling occurs without fertilisation in the integuments of the vrn2/+ emf2/+ mutant (as opposed to auxin signaling). Do the authors therefore propose the following: Auxin produced in the endosperm is transported to the integument where it inhibits PRC2 gene expression, which then in turn induce GA signaling/synthesis? This is not clear from the text, and as noted above needs to be addressed (perhaps a schematic model could clarify in addition to text in the Discussion).*

The reviewers are correct; we do propose that seed coat development is dependent on the auxin-driven removal of PRC2 function from the integuments, which induces GA signalling. We modified the text in the Discussion to make this clearer and included a model of the proposed mechanism in Figure 8, as suggested.

The repression of GA signaling by PRC2 is supported by the upregulation of the GA biosynthesis gene *GA20ox2*. We also observed downregulation of the GA catabolism genes *GA2ox2* and *GA2ox6* in *vrn2 emf2* ovules, which could be the consequence of increased expression of a repressor upon loss of PRC2 function. We included this information in the Results section and in Figure 6—figure supplement 1.

*2) In the last paragraph of the subsection “Auxin and sporophytic PRC2s work in the same pathway during seed coat development”, Figure 6: Although the link to auxin in this paper is strong, the link to GA is not well developed and the results shown here (looking at GFP:RGA in prc2 mutants) are rather weak. For example, is the expression of the pRGA-GFP:RGA transgene really equivalent at the RNA level in the two types of seeds shown? This should be tested. In addition, the prc2 mutant seeds shown have grown a lot compared to the wild-type by 5DAE, which can affect "quantitative" imaging, particularly in internal layers where the authors see the largest differences. The authors are not really comparing like with like in this experiment. Would it be possible to look earlier, when the seeds are at a more comparable developmental stage?*

The reviewers raise an important point, regarding the transcriptional regulation of *RGA* in the PRC2 mutant. According to our transcriptome analysis of the *vrn2 emf2/+* double mutant, *RGA* is in fact upregulated in the mutant when compared to WT ovules, meaning that decreased GFP:RGA signal in the PRC2 mutants is not due to decreased gene expression. We have added this information to the main text and to Figure 6—figure supplement 1.

We also included photos of *RGA::GFP:RGA* in the *swn clf* background at 2 DAE into Figure 6—figure supplement 2. In this case both WT and mutant ovules are approximately the same size, but the depletion of GFP:RGA can already be seen in the innermost layers of the mutant ovules.

*3) The experiments on the effects of auxin signaling vs. auxin production in the endosperm (Figure 3 and Figure 4) are elegant in their conception, but the interpretation of the data needs to be related to what has previously been published.*

*First, one cannot claim that seeds with a homozygous axr1/- axr1/- zygotic compartment are the same size as wild-type siblings just by showing a photograph (Figure 3). This should be quantified. Indeed, they look smaller (which would be consistent with the endosperm proliferation defects reported in the mutant in Figueiredo et al. 2015).*

As suggested by the reviewers, we quantified ovule and seed area before (2 DAE) and after fertilization (3 and 5 DAP) for WT and auxin signalling and biosynthesis mutants. We did not observe differences in ovule size prior to fertilization. However, at 3 DAP we observed that the auxin biosynthesis mutants produced smaller seeds when compared to WT and the signalling mutant, supporting our view that auxin biosynthesis in the endosperm rather than auxin signalling is required for seed coat initiation. At 5 DAP the signalling mutant had slightly smaller seeds compared to WT, but were still larger compared to the biosynthesis mutants. This data is now included in Figure 3—figure supplement 1.

Furthermore, our data and that of other groups indicates that endosperm proliferation per se does not impact on seed coat initiation (see below), thus, endosperm defects of the *axr1 axl-1* mutant should not directly affect the early stages of seed coat development. We have also modified our statements that auxin signalling is not necessary for full seed coat development, and now refer only to seed coat initiation.

*The argument that the defects in testa development in the auxin biosynthetic mutant backgrounds are due to endosperm-produced auxin is difficult to justify. This mutant will produce less auxin in the endosperm, but as pointed out in Figueiredo et al. 2015, this lack of auxin production causes major endosperm development problems and it is thus impossible to exclude the possibility that a lack of other endosperm-derived signals could underlie the serious defects in this background. A similar problem is associated with the DD65:TAA1; DD65:YUC1 lines. Figueiredo et al. (Figure 3) 2015 shows that these lines lead to the formation of an autonomous endosperm, thus again it is impossible to exclude an additional indirect effect of endosperm development as the reason for seed coat development. Thus, the conclusions from these experiments are perhaps overstated; although these experiments show that auxin production in the endosperm is needed for seed coat development, they do not prove that auxin itself is the signal. I.e., it is important to make any caveats clear and this will add to the value of your paper.*

While endosperm expansion has a mechanical role in supporting seed coat growth, the proliferation of the endosperm, per se, does not seem to have an impact in seed coat initiation. This is supported by the fact that mutants with severe endosperm proliferation defects, such as the *titan* mutants, fully develop a seed coat (Liu et al., 1998). Furthermore, ovules of transgenic lines expressing *DD65::TAA1; DD65::YUC6* initiate seed coat development before the central cell starts replicating. At 5 DAE the ovules are larger than WT and they stain strongly with vanillin (as we show in this manuscript), but autonomous central cell replication can only be seen at 6 DAE (Figueiredo et al., 2015). This suggests that endosperm proliferation is not required for seed coat initiation, if auxin is ectopically produced. Nevertheless, we agree that we cannot completely rule out that additional signals downstream of auxin have a role in seed coat development and have accounted for that option in the text.

*4) The authors argue that additional factors are required for PGP10 function, which is why the AGL62::PGP10 construct fails to complement the agl62 mutant. Although this is one possible explanation, it must follow from this experiment that these additional factors are also targets of AGL62 and therefore not present in the agl62 mutant. Perhaps it would be safer to reformulate with a more general statement that these results demonstrate that other targets of AGL62 are involved in facilitating auxin transport to the integuments. Some of these targets may indeed encode the additional factors required for PGP10 function.*

This hypothesis has now been included in the Discussion.

*5) In the last paragraph of the subsection “Auxin and GA trigger autonomous seed coat development”, it is stated that it is expected that the expression of hormone biosynthesis gene expression is specific to the integuments in BAN::GA3OX1 and not expressed in endosperm in the yuc6-2d line. Given the importance of this result, it would be useful if the authors could confirm this for instance by an in situ hybridisation using these lines.*

In Figure 2—figure supplement 1 we have included photos of a representative *BAN* reporter line developed in our lab, with the same promoter fragment used as in the *BAN::GA3ox1* construct. GFP activity can only be observed in the integuments, not in the gametophyte, in agreement with previously published data (Debeaujon et al. 2003). Ectopic expression of *YUC6* in the *yuc6-2d* line is conferred by the CaMV35S promoter that has been shown in several instances not to be expressed in the gametophytic cells, nor in the early endosperm (Boisnard-Lorig et al. (2001), Roszak and Köhler (2011)).

*6) In the second paragraph of the subsection “Auxin and sporophytic PRC2s work in the same pathway during seed coat development” and Figure 7: These results are over-interpreted. 1) The fact that transcriptional control of the PRC2-encoding genes is observed doesn't allow the authors to exclude regulation at the protein level. 2) More importantly, the fact that transcription of these genes decreases after pollination (there is a typo in the axis label of Figure 7), does not allow the authors to conclude that auxin-dependent removal of PRC2 function occurs at the level of transcription. The authors need to include either auxin treatments, or relevant mutant/transgenic lines in their Q-PCR experiment to allow them to draw this conclusion.*

We agree with the reviewers that, even though PCR2-coding genes are strongly downregulated following fertilization, this does not rule out an additional regulation of PRC2 at the protein level. We amended the text accordingly.

To test whether the downregulation of genes coding for PRC2 components was auxin-dependent, we analyzed their expression following auxin treatments in unfertilized ovules, which resulted in the downregulation of all genes tested, when compared to the mock control. This data was added to Figure 7—figure supplement 1.